# SINK OR SPIKE: REPRESENTATIONAL GEOMETRY MEDIATES GENERATION QUALITY IN LLMS

## ABSTRACT

Large Language Models (LLMs) rely on contextual representations for their input and output tokens in order to operate. Having an understanding of how these representations are organized and how this organization relates to the text the model generates is integral to the ongoing development of safer, more interpretable LLMs. However, even the representations created for a single prompt define a complicated, high-dimensional geometry, making this a difficult task. We take a step towards understanding this complicated relationship by using the singular value decomposition (SVD) of the representations for a single prompt to guide a perturbation procedure. We find two perturbation strategies that regularly cause two different LLMs to produce **coherent responses that are irrelevant to the prompt** or text that is **relevant but repetitive**, respectively. Analyzing the attention maps for these generations reveals a statistically significant trend where these two types of generations are associated with a negative or positive spike in the maximum attention value on the attention sink for each of the output tokens.

## 1 INTRODUCTION AND RELATED WORK

Large Language Models (LLMs) have had an enormous impact on several industries, academia, and even society at large due to their ability to create coherent, relevant text given a wide variety of prompts (Brown et al., 2020; Bin-Nashwan et al., 2023; Holmes & Tuomi, 2022; McElheran et al., 2024). LLMs produce text by creating contextual representations for each of the input tokens that depend on the representations of the tokens that precede them. The properties of these representations are paramount to determining the kind of text produced by the LLM (Park et al., 2023; Cancedda, 2024; Zhang et al., 2025). However, the representations' high dimensionality and the complicated process by which they are created makes it difficult to understand the relationship between an LLM's geometry and the text that it produces.

In this work, we attempt to understand this complicated relationship using the singular value decomposition (SVD) of the representations for a single prompt. We use these singular vectors to guide a perturbation procedure that projects out directions in representation space based on their explanatory power. Towards shedding light on how the complicated representational geometry of an LLM interplays with the generations and internal activations they produce, we contribute the following robust findings for both a Llama model (Dubey et al., 2024) and Mistral model (Jiang et al., 2024):

1. We demonstrate that keeping only a *Low-Band* of the spectrum produces **coherent responses that are irrelevant to the prompt**.

2. We show that removing a *Mid-Band* of the spectrum produces **relevant but repetitive** text.

3. Coherent, irrelevant responses and relevant, repetitive responses are associated with a statistically significant decrease or increase, respectively, in the attention values for the output tokens on the attention sink(s) (Xiao et al., 2023).

Several previous works have taken steps towards understanding the role of representational geometry in the operation of LLMs (Park et al., 2023; 2024; Sarfati et al., 2024; Kirsanov et al., 2025; Ruscio et al., 2025). Perhaps most similar to this paper are the work of Cancedda (2024) and Merullo et al. (2025). In both of these papers, the authors propose a means of relating the behaviour of LLMs to directions in representation space produced by SVD. In contrast to these works, we use the SVD of

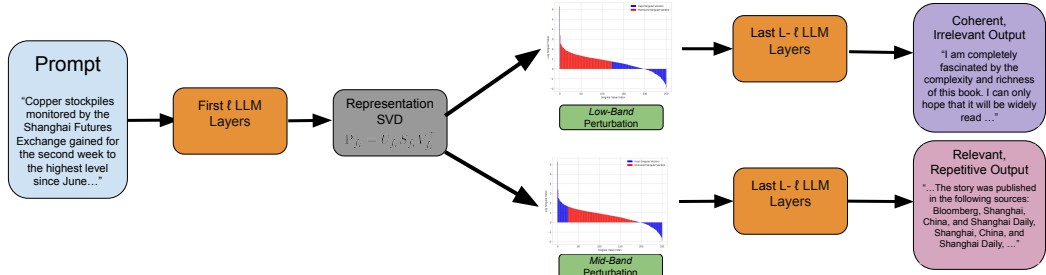

Figure 1: Our experimental pipeline and real examples of output text. Perturbing internal representations with specific SVD based projections yields generations with consistent characteristics.

the matrix of representations associated with each prompt separately, as this geometric object best describes the relative importance of each direction for a specific generation rather than in aggregate. We also apply these analyses to rigorously study free form text produced by LLMs.

## 2  METHODS

An LLM takes as input a tokenized prompt $p := \{x_i\}_{i=1}^S$ of length $S$, where each token satisfies $x_i \in \mathbb{R}^d$. The first layer of the network creates a set of contextual residual-stream representations, one for each token in the prompt: $p_{f_1} := \{f_1(x_i; x_i, \ldots, x_1)\}_{i=1}^S$, where $f_1$ represents the first layer of the network, $f_1(x_i; x_i, \ldots, x_1) \in \mathbb{R}^d$. The representation for the token $x_i$ will always be conditioned on previous tokens, and so we denote it simply as $f_1(x_i)$. Subsequent layers $f_\ell$ produce their residual-stream representations using the output of the previous layer: $p_{f_\ell} := \{f_\ell(f_{\ell-1}(x_i))\}_{i=1}^S$. These representations cascade through the model until they are finally used with the readout head to sample a new token $x_{S+1}$, which is then fed through the model to produce $x_{S+2}$, until a full new set of output tokens $o := \{x_i\}_{i=S+1}^{S+O}$ has been produced, each with corresponding representations at each layer.

### 2.1  PER-PROMPT REPRESENTATION SPECTRA AND PROJECTION

A common way of characterizing the geometry of a pool of vectors is using the SVD of the matrix containing those vectors. Specifically, we can stack the vectors in $p_{f_\ell}$ into a matrix $P_{f_\ell} \in \mathbb{R}^{S \times D}$. SVD then decomposes the matrix into a product of three matrices: $P_{f_\ell} = U_{f_\ell} S_{f_\ell} V_{f_\ell}^\top$. Most important for understanding our analysis are the matrices $V_{f_\ell}^\top$, which has rows representing directions in representation space, and $S_{f_\ell}$, which is a diagonal matrix with values representing how much of the representations' magnitude is along the corresponding direction in $V_{f_\ell}^\top$.

Seeing as SVD gives us an understanding of the directions in the representation space encompassing most of the magnitude for representations from a prompt, we use this as a means of examining an individual prompt's geometry. Specifically, we study the impact of ablating selected singular vectors from the representation space on the text generated by the model. Consider some subset of the rows of $V_{f_\ell}^\top$, $\Omega \subseteq Rows(V_{f_\ell}^\top)$: we can use this set to create a new version of $p_{f_\ell}$ where the vectors in $\Omega$ are projected out (details in Appendix A.1):

$$p_{f_\ell, \Omega} := \{f_\ell(f_{\ell-1}(x_i)) - \sum_{e \in \Omega} <e, f_\ell(f_{\ell-1}(x_i))>\}_{i=1}^S. \tag{1}$$

These perturbed representations can be passed through the rest of the layers of the model to produce a new generation, potentially different from the unperturbed version. To select vectors for projection, we use the quantiles of the associated singular values. Specifically, for layer index $\ell$ and fractions $\alpha, \beta \in [0, 1]$, $\Omega_{\ell, \alpha, \beta}$ defines the set of singular vectors from layer $\ell$ that have singular values greater than $\alpha$ of the singular values and less than $\beta$ of the singular values. We use these spectral bands to understand what directions in representation space of differing magnitudes account for in the operation of the model.

## 2.2 GENERATION CRITERIA AND AUTOMATIC EVALUATION

We will be using different perturbation sets and determining the kinds of text generations they tend to produce. To do this we choose three criteria, which we choose to cast as binary:

- **Coherence**: Whether the response is composed of readable text following some kind of grammar conventions, rather than randomly selected words, or characters.
- **Relevance**: Whether the subject of the response is related to the subject of the prompt.
- **Repetitiveness**: Whether the response is composed of repeated sentences, sentence fragments, words or individual characters.

Across the thousands of generations we produce, we use an automatic evaluation set up that leverages GPT5-Mini (Singh et al., 2025) as well as Llama-3.1-70B-Instruct (Dubey et al., 2024) to evaluate whether the response meets these criteria. This approach is common but can produce errors (Gu et al., 2024; Szymanski et al., 2025). However, our qualitative judgement of these evaluations found them to be reliable. Details regarding this procedure are in Appendix A.2.

## 3 RESULTS

We now examine how using different perturbations can effect the outputs of an LLM. We experiment with both Llama-3.1-8B-Instruct (Dubey et al., 2024) and Mistral-7B-Instruct-v0.3 (Jiang et al., 2024). We sample two random prompts from each of four different datasets (see Appendix A.3) and generate 10 responses from each model using a variety of perturbations (see Appendix A.4). We quantify how these perturbations influence the generated text, and study the associated attention patterns to determine how the induced attention maps yield different kinds of generated text.

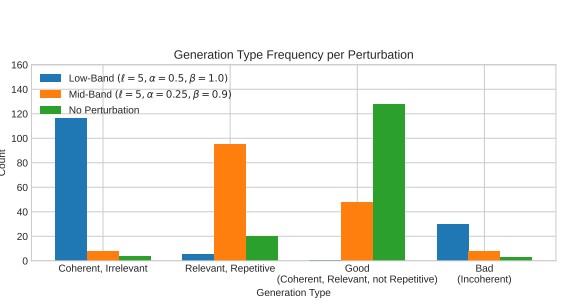
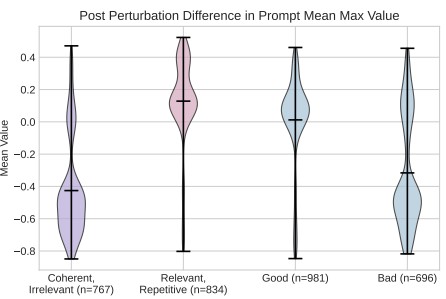

(a) Number of responses of each type described in Section 3.1 for both Llama and Mistral across three perturbation regimes.

(b) Differences in mean max attention value in **Prompt** before and after perturbation for the four generation types described in Section 3.1.

Figure 2: Analyses for different response types.

## 3.1 PERTURBATIONS CONTROLLABLY INFLUENCE GENERATION CHARACTERISTICS

We focus on two types of perturbations in particular: *Low-Band* removes all singular vectors between the $50^{th}$ and $100^{th}$ percentiles of singular values (with the exception of the top singular vector) in the $5^{th}$ layer ($\Omega_{5,0.5,1.0}$), and *Mid-Band* removes all singular vectors between the $25^{th}$ and $90^{th}$ percentiles of singular values in the $5^{th}$ layer ($\Omega_{5,0.25,0.9}$).

Across multiple generations and multiple prompts, *Low-Band* tends to produce **coherent, but irrelevant text**, while *Mid-Band* tends to produce **relevant, but repetitive text**. Figure 2a shows that *Low-Band* produces more coherent, irrelevant responses than any other kind of response type, and it produces these responses more than either of our other generation regimes. Similarly, *Mid-Band* produces relevant, repetitive responses in a similar manner. As a control, we also show that unperturbed models produce relevant, coherent, and non-repetitive responses (*Good*) most frequently. Finally, these perturbations have a controlled impact: both *Low-Band* and *Mid-Band* produce a specific type

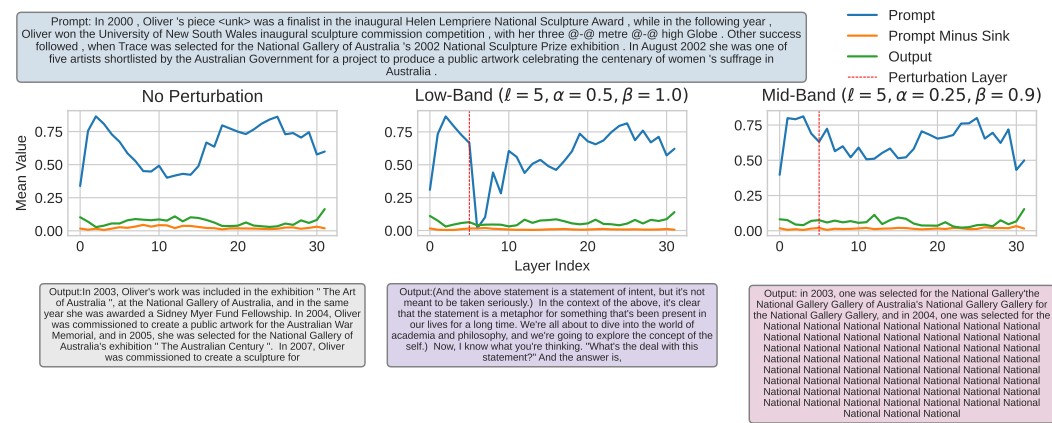

Figure 3: Attention pattern statistics (Section 3.2) across layers for three perturbation regimes and their associated outputs.

of generation more than any other kind of generation, particular more than corrupt generations that are neither relevant nor coherent, or relevant but not coherent or repetitive (*Bad*).

## 3.2 PERTURBATIONS CONTROLLABLY INFLUENCE ATTENTION PATTERNS

We next investigate how perturbations effect the model's attention maps, and how those attention maps relate to different generation types. We study the following regions of the attention maps:

- **Prompt**: The attention values for output tokens on tokens from the prompt.
- **Prompt Minus Sink**: The same as **Prompt**, **excluding** the attention sink token(s).
- **Output**: The attention values for output tokens on other output tokens.

We compute the mean maximum attention value for each output token on each population. We also take the difference of this average value for the **Prompt** population before and after perturbation. More details can be found in Appendix A.5.

Figure 2b demonstrates that this statistic is substantially different across the four types of responses described in Section 3.1. Coherent, irrelevant responses show a dramatic drop in mean max attention in **Prompt**, while relevant, repetitive responses show a slight rise. Most importantly, the difference in means between coherent, irrelevant responses and relevant, repetitive responses with the rest of the generations is statistically significant according to a Welch's t-test ($p = 1.5 * 10^{-151}$ and $p = 8.9 * 10^{-157}$, respectively). Additional results in Appendix A.5 show that the same trend is not present for the **Prompt Minus Sink** population, showing this drop is primarily attributable to the attention sink. This demonstrates that perturbing per-prompt geometry is able to have a reliable effect on the generated text by consistently manipulating the model's internal attention patterns.

As demonstrated in Figure 3, these statistics seem to have specific patterns for the two perturbation types. For *Low-Band*, the **Prompt** max attention value sinks dramatically, while the same isn't true for the max value for **Prompt Minus Sink**, indicating that the attention sink value has dropped significantly. Meanwhile, for *Mid-Band* we see a slight rise in the max **Prompt** attention value, indicating that the attention sink is getting a higher maximum value after the perturbation.

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

## A  APPENDIX

### A.1  PROJECTION DETAILS

We start the projection procedure by organizing the vectors in $p_{f_\ell}$ into a matrix $P_{f_\ell} \in \mathbb{R}^{S \times D}$. SVD then decomposes the matrix into a product of three matrices: $P_{f_\ell} = U_{f_\ell} S_{f_\ell} V_{f_\ell}^\top$. Consider some subset of the rows of $V_{f_\ell}^\top$, $\Omega \subseteq Rows(V_{f_\ell}^\top)$: we can use this set to create a new version of $p_{f_\ell}$ where the vectors in $\Omega$ are projected out:

$$p_{f_\ell, \Omega} := \{f_\ell(f_{\ell-1}(x_i)) - \sum_{e \in \Omega} < e, f_\ell(f_{\ell-1}(x_i)) >\}_{i=1}^S. \tag{2}$$

These new representations are fed through the rest of the model to decode a new token $x_{S+1}$. When this new token's representation reaches layer $\ell$, we project out the **fixed** singular vectors in $\Omega$ from $f_\ell(f_{\ell-1}(x_{S+1}))$ and send that projected representation through the rest of the network. This is repeated for every output token in the decoding process.

### A.2  AUTOEVAL PROCEDURE

We treat coherence, relevance, and repetitiveness as binary attributes of a response given a prompt. For the first two attributes we use GPT5-Mini, and for the last we use Llama-3.1-70B-Instruct as the task is quite straightforward. Evaluating for each of these attributes is accomplished using a custom in-context-learning prompt for each of the tasks that prompts the model to provide a binary score after the word "score". If a binary score is not matched this way, we give a score of "N/A" and this example is omit from our analysis.

### A.3  DATASETS AND PROMPT SELECTION

We select four different datasets across four domains: the TweetEval dataset (Barbieri et al., 2018; 2020), the Financial News Dataset from Bloomberg and Reuters (Ding et al., 2014; Benayoun, 2024; Philippe Remy, 2015), the WikiText dataset (Merity et al., 2016), and a dataset of paragraphs from the Lord of the Rings fantasy series[1]. For each of these, we randomly sample two prompts meeting some minimum length.

### A.4  LIST OF PERTURBATIONS

We produce generations for all 8 prompts across both models for the following perturbation regimes:

- $\alpha = 0.5, \beta = 1.0, \ell \in \{5, 12, 15, 25\}$ (first vector not removed),
- $\alpha = 0.75, \beta = 1.0, \ell \in \{5, 12, 15, 25\}$ (first vector not removed),
- $\alpha = 0.25, \beta = 0.9, \ell \in \{5, 12, 15, 25\}$,
- $\alpha = 0.0, \beta = 0.95, \ell \in \{5, 12, 15, 25\}$,
- $\alpha = 0.25, \beta = 0.95, \ell \in \{5, 12, 15, 25\}$.

### A.5  ATTENTION STATISTIC CALCULATION

For a given prompt + generation $p$ with $S + O$ tokens, there is a $(S + O) \times (S + O)$ attention matrix for each attention head in each layer. Based on that, we focus on the $O \times (S + O)$ submatrix corresponding just to the attention values for the output tokens. This can be further partitioned into the attention values corresponding to our **Prompt**, **Prompt Minus Sink**, and **Output** regions. The first $S$ columns corresponds to **Prompt**, columns $k = 1, 3$ to $S$ correspond to **Prompt Minus Sink** ($k = 1$ for Llama, $k = 3$ for Mistral), and the last $O$ columns corresponds to **Output**. For each of these submatrices, we take the maximum value in each row, and take the average of those maximum values. We then take average of these values across the attention heads, and there is one

---

[1]https://huggingface.co/datasets/AL49/lotr_paragraphs

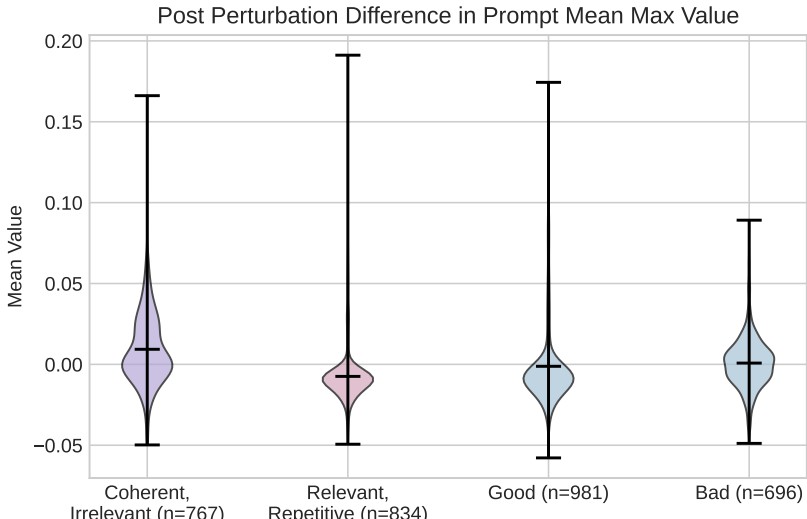

Figure 4: Differences in mean max attention value in **Prompt Minus Sink** before and after perturbation for the four generation types described in Section 3.1. Notably, the dramatic rises and drops seen in Figure 2b in these values are not present.

such value for each layer for each prompt. These are plotted for each layer in Figure 3. In Figure 2b, we are plotting the distribution of differences between these values for the layers before and after a perturbation.

In Figure 4, we see that calculating the differences in mean max attention value in **Prompt Minus Sink** before and after perturbation does not yield a precipitous drop for coherent, irrelevant responses nor a slight rise for relevant, repetitive responses. Given that this region of the attention map differs from **Prompt** only in the exclusion of the attention sink, it stands to reason that the attention sink is responsible for the trends seen in Figure 2b.

