# OpenReview forum: "Sink or Spike: Representational Geometry Mediates Generation Quality in LLMs"
_ICLR.cc/2026/Workshop/Sci4DL — Submitted to Sci4DL 2026_

### Official Review · Reviewer_zgvK · 2026-02-08

**Fit:** 3
**Significance:** 2
**Confidence:** 2

**Summary:**

The paper investigates how the high-dimensional geometry of LLMs representations influences text generation quality. By applying Singular Value Decomposition (SVD) to representations of specific prompts, authors identify spectral bands of singular vectors that correlate with distinct failure modes: Removing the low-band of singular values tends to produce coherent but irrelevant responses, while removing a mid-band tends to result in relevant but repetitive text. They connect these geometric perturbations to changes in attention maps, showing that irrelevant or repetitive generations are linked to statistical decreases or increases in attention sink values.

**Strengths:**

The paper has different strengths:
- The core scientific question is interesting and compelling, and correctly methodologically addressed with the SVD analysis.
- The per-prompt SVD decomposition provides a highly granular view to analyze how specific magnitude-based directions in the representation space affect model behaviour.
- The paper successfully identifies a statistical link between the internal geometric properties and the attention sink mechanism, moving beyond purely qualitative observations.

**Suggestions:**

There are also a few recommendations/improvements I would like to share with the authors.
First, regarding the paper writing:
- There is no formal conclusion section. While page limits are strict, a concise summary of the findings and their implications is necessary.
- Appendix A.1 is highly repetitive, providing little information not already present in the main text; specifically, Equation 2 appears to be identical to Equation 1.

Regarding scientific work, also a few suggestions:
- The current approach of projecting out, i.e., completely removing some directions, is an extreme intervention. It would be scientifically interesting to analyze soft perturbations, i.e. reducing the singular values by a certain percentage rather than total removal, to see if the failure modes emerge gradually.
- Results are currently focused just on layer 5, but Appendix A.4 suggests that data were collected for different layers. It is necessary to include these to understand if the trends are layer-dependent. Additionally, exploring the effect of simultaneous perturbations across multiple-layer combinations would be a significant extension.
- The current low-band and mid-band percentiles overlap. Authors should justify this overlap or consider a clean tripartite division (such as <33rd, 33rd–66th, >66th percentiles). Also, an analysis of the high-band is missing and would serve as a critical baseline together with the no-intervention.
- The authors mention they remove the low-band "with the exception of the top singular vector". Clarifying the scientific rationale for protecting this specific vector would strengthen the methodological depth. Also, I would like to see an ablation of just removing the top singular vector/removing all except the top singular vector.
- While using two models is enough for a workshop, the results should be more clearly disaggregated. In Figure 2b, the violin plots clearly show distinct bimodal distributions for some of the analyzed generation types. The authors should clarify whether these clusters correspond to the two different models or the four diverse datasets. Also, does dataset domain (creative vs. factual) influence the prominence of specific failure modes?
- Appendix A.2 lacks the specific prompt templates used for the LLM-as-a-judge evaluation. For a scientific workshop, providing the exact prompts is essential for reproducibility.

---

### Official Review · Reviewer_1JWa · 2026-02-25

**Fit:** 3
**Significance:** 2
**Confidence:** 2

**Summary:**

This paper investigates the effect of representational geometry of LLMs on generation quality. Concretely, the paper computes per-prompt SVD of representations from the residual steam at specific layers and perturbs the representations by projecting out selected spectral bands. The paper observes two interesting phenomena: removing low-band SVD components produces coherent but irrelevant text, and removing mid-band components produces relevant but repetitive text. Further, the paper shows that these generation types correspond to systematic decrease / increase in attention to prompts and attention sinks.

**Strengths:**

- The findings are consistent across models from different families (Llama and Qwen) and the SVD decomposition and perturbation is a simple interpretability tool to evaluate these behaviors in generated responses.

**Suggestions:**

- The set of prompts is small (2 prompts each for 4 datasets).
- The band selection parameters seem heuristic (why is the mid band from 0.25 to 0.9 quantile?).
- Since the set of prompts and responses is small, a qualitative human evaluation of all responses is feasible and encouraged.

---

### Official Review · Reviewer_diKm · 2026-02-25

**Fit:** 2
**Significance:** 2
**Confidence:** 2

**Summary:**

The authors apply SVD-guided perturbations to LLM representations to link representational geometry with generation quality. In Llama and Mistral, keeping Low-Band vectors causes prompt-irrelevant output, while removing Mid-Band vectors produces repetitive text. These patterns correlate significantly with attention sink activation shifts.

**Strengths:**

- Mechanistic Insights: The work provides a compelling bridge between internal geometric structures and specific model behaviors like looping or loss of context.
- Generalizability: The findings are consistent across two different model families, Llama-3.1-8B and Mistral-7B-Instruct-v0.3, suggesting these are potentially fundamental properties.
- Statistical Foundation: The reported statistical significance is very high, at least for this small scale evaluation.

**Suggestions:**

- Linear Methodology: The paper relies on SVD, which is a linear method, to characterize a geometry that the authors themselves describe as complicated. Is it possible that nonlinear geometric properties also play a role in mediation that SVD cannot capture?
- Title and Scope: The title "Representational Geometry Mediates Generation Quality" feels a bit broad. Given the focus is specifically on repetition and prompt-relevance failures via spectral band ablation, "quality" as a whole may be over-generalized. Similar for "representational geometry", as only one (linear) characteristic of this geometry is explored.
- Sample Size: While the statistical significance is strong, the actual evaluation relied on generating only 10 responses per model/perturbation for a small set of 8 prompts. I'm curious to see this scale up, including testing on more models, datasets, etc. Moreover, how else do the authors think that mathematically-grounded quantitative measures of prompt representation geometry could relate to qualitative behavioral trends? Is this a promising research direction?
- Appendix Expansion: Including more qualitative prompt-output triplets in the appendix, similar to the examples in Figure 3, would help readers better visualize the differences between the identified failure modes.

---

### Meta-Review · Area_Chair_oDWr · 2026-03-01

**Recommendation:** Reject

**Metareview:**

Recommending rejection, due to the absence of a clear scientific hypothesis and relatively weak and underpowered experiments. I encourage the authors to work on this, and the reviewer suggestions, in potential re-submissions.

---

### Decision · Program_Chairs · 2026-03-02

Reject